# Soldier Load Carriage, Injuries, Rehabilitation and Physical Conditioning: An International Approach

**DOI:** 10.3390/ijerph18084010

**Published:** 2021-04-11

**Authors:** Robin Orr, Rodney Pope, Thiago Jambo Alves Lopes, Dieter Leyk, Sam Blacker, Beatriz Sanz Bustillo-Aguirre, Joseph J. Knapik

**Affiliations:** 1Tactical Research Unit, Bond University, Gold Coast 4213, Australia; rpope@csu.edu.au (R.P.); joseph.j.knapik.civ@mail.mil (J.J.K.); 2School of Community Health, Charles Sturt University, Albury 2640, Australia; 3Research Laboratory of Exercise Science, Centro de Educação Física Almirante Adalberto Nunes, Brazilian Navy, Rio de Janeiro 21941-901, Brazil; ThiagoJambo@hotmail.com; 4Post-Graduation Program in Operational Human Performance/PPGDHO, Brazilian Air Force, University of the Air Force, Rio de Janeiro 21941-901, Brazil; 5Research Group Epidemiology of Performance, German Sport University Cologne, 50933 Cologne, Germany; dieterleyk@bundeswehr.org; 6Bundeswehr Institute for Preventive Medicine, 56626 Andernach, Germany; 7Occupational Performance Research Group, Institute of Sport, University of Chichester, West Sussex PO19 6PE, UK; S.Blacker@chi.ac.uk; 8Ministry of Defence, Paseo de la Castellana 109, 28046 Madrid, Spain; beabustillo@telefonica.net; 9Universidad San Pablo-CEU, CEU Universities, Avenida Montepríncipe s/n, Bohadilla del Monte, 28668 Madrid, Spain; 10United States Army Research Institute of Environmental Medicine, Natick, MA 01760, USA

**Keywords:** military, exposure, occupational injury, pack march, reconditioning, return to work, injury risk management

## Abstract

Soldiers are often required to carry heavy loads that can exceed 45 kg. The physiological costs and biomechanical responses to these loads, whilst varying with the contexts in which they are carried, have led to soldier injuries. These injuries can range from musculoskeletal injuries (e.g., joint/ligamentous injuries and stress fractures) to neurological injuries (e.g., paresthesias), and impact on both the soldier and the army in which they serve. Following treatment to facilitate initial recovery from injuries, soldiers must be progressively reconditioned for load carriage. Optimal conditioning and reconditioning practices include load carriage sessions with a frequency of one session every 10–14 days in conjunction with a program of both resistance and aerobic training. Speed of march and grade and type of terrain covered are factors that can be adjusted to manipulate load carriage intensity, limiting the need to adjust load weight alone. Factors external to the load carriage program, such as other military duties, can also impart physical loading and must be considered as part of any load carriage conditioning/reconditioning program.

## 1. Introduction

From the deserts in Africa [1] and Iraq [2] to the jungles of Vietnam [3]; from the marshes of the Falklands [4] to the urban sprawl of Somalia [5]; and from traversing the flat lands of rice paddies in Vietnam [6] and poppy fields in Afghanistan [7] to the hilly Kokoda Track of Papua New Guinea [8], Toktong pass of Korea [9] and Shah-i-Kot Valley of Afghanistan [10], soldiers have been required to fight battles for survival while wearing and carrying heavy occupational loads. These loads are composed of equipment and stores designed to ensure protection (e.g., body armour), lethality (e.g., weapon systems and ammunition), and sustainment (e.g., food and water) and are worn on the head (e.g., ballistic helmet), torso (e.g., body armour, load bearing vests, chest rigs, patrol pack), thigh (e.g., side arm or stores in thigh pockets) and feet (e.g., boots) as well as being carried in the hands (e.g., primary weapon systems). Whilst these items are important for soldier survival, the loads also impart risks due to factors which include fatigue and the increased energy cost associated with prolonged load carriage [11,12]. The loads carried by the soldier may reduce their physiological capability, mobility, attention to task, marksmanship and grenade throw ability [13,14,15,16,17,18,19]. Furthermore, load carriage tasks have been associated with soldier injuries and a history of load carriage injuries increases the risk of future load carriage injuries [20,21]. Hence, preventing load carriage injuries and optimising the physical rehabilitation of injured soldiers to reduce future injury risks and enhance physical, cognitive and technical performance is of importance.

The purpose of this article is to review the physiological and biomechanical responses to load carriage and how these relate to injuries, injury prevention, and rehabilitation after injury.

## 2. The Weight of the Soldier’s Load

It must be acknowledged that the requirement to carry load is not unique to soldiers. Law enforcement personnel are required to carry daily occupational loads that can range from approximately 10 kg for a general duties officer [22] up to approximately 20–25 kg [23,24] and even 40 kg [25] for a specialist tactical response officer. Likewise, firefighters are often required to generally carry loads of over 20 kg while wearing firefighting personal protective clothing [26]. Nevertheless, although these loads are notable, they are typically lighter than those carried by soldiers as soldiers are often in situations where they do not have the relatively close support networks that are commonly available to other tactical groups such as law enforcement and firefighting personnel. As such, soldiers must carry more equipment and stores. While the soldier’s load may vary depending on their Corps (e.g., infantry, armoured, and artillery [27]), roles within a given unit (e.g., rifleman, grenadier, and section commander [28]) and tasks (e.g., patrolling on foot or in a vehicle, and sentry duty [27]), the loads carried by US [21,28,29,30], Australian [27], British [31], Spanish [18], and German [32,33,34] soldiers can weigh from approximately 25 kg to well over 45 kg. Of most concern, research suggests that regardless of advances in weaponry, changes in theatres of war, and changes to technology in general, the loads carried by soldiers into combat are increasing [15,35]. Not only may the weight of these loads differ between occupations but the contexts in which these loads are carried may vary, law enforcement officers may wear their relatively lighter loads daily for the duration of their career, while firefighters will don their loads for specific periods (e.g., while fighting a structural fire) and military personnel, again depending on their unit’s roles and tasks, could carry loads for varying lengths of time and over varying distances [36,37].

## 3. Physiological Responses to Load Carriage

The weight of the loads carried by soldiers is known to elicit a physiological cost. Increases in load weight have been found to reduce endurance time [38] and increase the energy cost of walking (forwards and backwards, and up and down stairs), and running [39,40,41]. However, it is not only the weight of the load being carried that influences the soldier’s physiological responses to a load carriage task, so too do the contexts in which loads are carried [15]. An example of a contextual influence is the speed at which the carrier is moving while the load is carried, whereby increases in speed of movement increase the energy cost of carrying a given load to the point where it has been suggested that increases in speed may have a greater impact on energy expenditure than increases in load weight [42].

Changes in both terrain gradients and terrain surfaces have also been observed to affect the energy costs of load carriage, such that increases in gradients traversed increase the energy costs of carrying loads [43,44,45]. As with speed, increases in the grade (angle) of the terrain may impart a greater energy cost to the soldier than increases in load weight [45]. When considering the incline of terrain, Crowder et al. [45] advised that the grade might be more important than the load from an energy cost perspective because a 1% increase in grade increased energy cost by approximately 10-fold more than a 1% increase in load with no change in grade. Changes in energy cost per unit of downhill gradient are not linear but rather assume a “U” shape. That is, as the downhill grade increases, energy cost initially decreases, reaches a minimum and then increases again. Lloyd and Cooke [46] observed that as gradients declined from zero to −5%, there was a minor reduction in oxygen consumption, with consumption then increasing steadily as the gradient decreased from −12 to −27%. Further, Santee et al. [47], investigating the impact of declining gradients of up to −12% (during 20 min of treadmill walking carrying 9.1 and 18.1 kg backpacks), reported similar reductions in energy requirements occurring with declines in grade to −12%, when compared to level walking. Considering these findings regarding the impacts of incline and decline terrain gradients on load carriage energy expenditure, it is noteworthy that studies assessing energy costs on both inclining (up to +27%) and declining gradients (down to −30%) have reported that load carriage on inclining gradients is more energy costly than that on decline gradients [46,47,48]. The nature of the terrain to be traversed should also be considered, as different surfaces incur different physiological costs. Energy cost when carrying load increases across the following list of terrain types: sealed roads, dirt roads, light bush, heavy bush, swamp, loose sand, and deep (10–20 cm) snow [49,50].

As such, not only must the weight being carried by the soldier be considered but so too must the context in which the load is carried, with speed of march and grade and type of terrain also requiring consideration. These contextual factors may provide a means for increasing the load carriage training intensity for a soldier recovering from injury without increasing their carried weight, and so may be of value in a load carriage reconditioning program.

## 4. Biomechanical Responses to Load Carriage

In addition to the physiological costs, load carriage tasks alter biomechanics, including changes to the soldier’s posture, gait kinematics (stride length, stride frequency, etc.), and ground reaction forces when walking [15,51]. Alterations to forward trunk lean, spine shape, spinal compression, spinal shearing forces, and thoraco-pelvic rhythm have all been associated with carrying loads [52,53,54,55]. Load carriage increases postural sway [56,57,58,59,60] and the amount of force generated in the medial–lateral axis [61]. Likewise, load carriage has been associated with changes in the parameters of gait, including changes in the duration of the double support phase, stride length and stride frequency, with these changes dependent on the load and the sex of the carrier [51,55,62,63,64,65,66,67]. Finally, ground reaction forces increase in downward, antero-posterior, and medio-lateral directions as the carried load weight increases [61,62,63,67,68,69,70].

## 5. Injuries Associated with Soldier Load Carriage

When the aforementioned physiological and biomechanical factors are considered in combination, the potential for soldier load carriage to lead to injuries becomes apparent. The physiological cost of carrying the load can lead to fatigue [12]. The increased energy cost and increased and repetitive muscular force requirements can lead to central or generalised fatigue and localised muscle fatigue increasing the risk of injury to the soldier [11]. Changes to spinal loading, gait patterns, and ground impact forces (through increases in the total volume of impact forces over time [61]) can likely increase the risk of musculoskeletal injuries, particularly during prolonged and/or high-intensity load carriage activities [71].

In terms of injury types, load carriage tasks are associated with causing injuries in soldiers that range from fractures to ligamentous damage and from skin blistering to neurological injuries [15,20,21,30,72,73,74,75,76]. Injury body site data from both specific load carriage events [69,77] and longitudinal studies [20,74] suggest that the lower limbs are a leading site of injury from load carriage, with the knee, ankle and foot found to be common sites of musculoskeletal injuries [69,77]. However, some differences in the distribution of these injury sites between sexes may exist. A study by Orr et al. [78] investigating sex differences in soldier load carriage injuries found that female soldiers were more than twice as likely as their male counterparts to suffer from a foot injury, whilst in male soldiers, the ankle was the leading lower-limb injury site and accounted for a larger proportion of injuries. All other injury anatomical locations were similar for men and women, in the proportions of injuries they hosted.

In aggregated injury data representing all injuries in an army population accrued over an extended period of time, the back is typically, although not always [73], the second most common site of load carriage injuries after the lower limbs [20,69]. However, in a study of a single load carriage event, the back was the leading site of injuries that led to a soldier’s inability to complete the march [69]. This finding is supported by a retrospective cohort study by Orr et al. [78], which found that, when injuries were separated into body sites (e.g., knee, ankle, and foot), the lower back presented as the leading site of injuries in both male and female soldiers [78]. Again, a sex-specific difference was found, in which female soldiers, while suffering similar proportions of lower back injuries, suffered more severe lower back injuries than male soldiers [78]. When an individual walks with a backpack load, forward lean is increased, generating cyclic stresses on the vertebrae, intervertebral discs, muscles, and other spinal structures with each step [21]. Further, heavy loads do not move in synchrony with the trunk and trunk stiffness increases with such heavy loads due to both active co-contraction of the abdominal and paraspinal muscles and paraspinal reflexes. The combined stresses on vertebra, discs, muscles and other spinal structures are likely associated with back pain and injuries experienced in susceptible individuals [21].

Stress fractures (fatigue fractures) appear to be due to a bone remodeling imbalance, where the physiological processes that remove bone tissue outpace processes that produce new bone in stressed areas; stress fractures are typically associated with repetitive bone loading in activities such as walking with loads and other repetitive actions for which individuals are not adequately conditioned [79,80]. In military populations, common sites of stress fractures include the pelvis, tibia, calcaneus, and metatarsals [81,82]. Acknowledging that factors other than load carriage (e.g., running volume [30]) contribute to stress fractures, load carriage itself has been found to be a cause of stress fractures. In fact, the first report of stress fractures in the literature was by Dr Breithaupt, who noted the condition in Prussian Army soldiers returning from long marches, although he likely misconstrued the aetiology [83]. The injury was later termed “march fracture” since it was often seen in soldiers involved in marching with loads [83,84]. In a study of pelvic stress fractures in female soldiers, Pope et al. [81] identified the longer step length requirement typical of female soldiers (to keep ‘in-step’ with male soldiers, who are on average taller, during pack marches) as contributing to specific pubic ramus stress fractures in the female soldiers.

Neurological injuries are also associated with load carriage tasks and include several paresthesias (brachial plexus, digitalgia and meralgia) [75,76]. With a mechanism of injury that can involve either neural traction or compression, causes of paresthesias include loads transferred through backpack shoulder straps (brachial plexus palsy [76]), poorly fitting boots (digitalgia paresthetica [76]) or wearing body armour that compresses the thighs while seated for long periods of time (meralgia paresthetica [75]). Although the incidence rates for these injuries are not high compared to those for other load carriage injuries [75,76], recovery can take up to several months [85], with surgical intervention for brachial plexus palsy recommended if there is failure to recover strength and endurance after 24 months [86].

### The Wider Impacts of Soldier Load Carriage Injuries

The impacts of injuries to soldiers while carrying loads can be traced through history. Circa 400 BC, following the long marches of Cyrus’ infamous 10,000 Greek mercenaries, the army was thought to have suffered from stress fractures, torn ligaments, muscle damage, blisters and abrasions [87]. In 1870, the Prussian Guards fighting in the Franco-Prussian War left the Rhine with 30,000 soldiers but lost 12,000 fighting soldiers from fatigue induced by carrying heavy loads over the weeks of marching [88]. In 1944, during World War II, American troops were said to have been so overloaded that their loads were attributed with causing deaths in the water during landings at Omaha Beach [89]. In 1983, US soldiers assaulting an airhead in Grenada were so overloaded that a large number of them were left on the roadside on intravenous drips and unable to continue to fight [89]. More recently, research suggested that nearly a half (45%) of US combat forces reported suffering a musculoskeletal injury during a 12-month deployment and the authors concluded that tasks requiring physical energy expenditure such as load carriage, lifting, or standing resulted in an increased risk of musculoskeletal injury in this study; lifting/carrying, dismounted patrols, and physical training were associated with 26% of the reported musculoskeletal injuries [90].

Thus, the potential impact of load carriage injuries on an army can be devastating, adversely affecting its combat capability [91]. Noting the impacts of injuries on army capability, injury prevention, including prevention of recurrences through rehabilitation, is considered by some as a force multiplier [92], a term acknowledging that reducing injuries and optimising rehabilitation will substantially increase combat capability. This makes sense intuitively. For example, if a soldier is injured and they cannot go out on a foot patrol, the rest of the unit must then perform the task with fewer personnel, who may now have to carry additional mission essential stores (increasing their load) with a reduced lethality capability (one less soldier to engage the enemy), factors that both increase the risk of injury and death to the remaining soldiers.

Noting these impacts of load carriage injuries on strategic capability, preventing injuries is of critical importance. This assertion is strengthened by the fact that previous injury is a risk factor for future injury [20,93,94,95]. A study by Orr et al. [20] found that, of soldiers injured while carrying load during basic training, 32% sustained an additional injury (to the same or another body site) within the first 12 months of service in an operational unit and overall 52% of those injured reported sustaining an additional load carriage injury (to the same or another body site) at some time during their career. Not only is the prevention of load carriage injuries paramount, but so too is the optimisation of an injured soldier’s rehabilitation for return to work (especially after a load carriage injury), as their ability to return affects not only the individual soldier but also the fighting capability of their unit.

## 6. Physical Conditioning and Rehabilitation for Soldier Load Carriage Tasks

Soldiers will often be the physical platform upon which heavy loads are carried [96], especially in austere or isolated environments where alternative methods are not readily available. In attempting to reduce injury risk in load carriage contexts, the hierarchy of controls for hazards should be considered. This hierarchy includes elimination, substitution, isolation, engineering controls, administrative controls, and personal protective equipment [96]. Elimination, substitution, isolation, and significant engineering controls are rarely considered viable options for control of injury risks arising from load carriage in military training and operational settings, due to requirements such as stealth, movement (affecting proximity of stores), and self-sufficiency during operational duties. The soldier must therefore be physically robust enough to withstand the forces imparted by the loads they must carry [96]. Recognition of the need for soldiers to be physically conditioned and robust enough to carry load is not new and can be traced back to Flavius Renatus who, in his epitome *Epitoma rei militaris*, described the physical training of Roman soldiers to carry loads and march long distances [97]. Following injury from any cause, soldiers must therefore be supported to recover, return to work and be reconditioned for load carriage.

Immediately following injury, emphasis will necessarily be on early detection, stabilisation, diagnosis and treatment of the injuries, to prevent further harm [98]. These topics are beyond the scope of this article and clinical guidance in these areas is available from many other sources, dependent on the specific type of injury. However, for optimal outcomes, immediately following injury stabilisation the focus must turn to: preventing unnecessary loss of physical conditioning; returning the soldier to work as soon as possible using approaches that will support their recovery from injury and build their confidence in their capacity to return to work; and facilitating the progressive physical reconditioning of the soldier for load carriage and other military tasks [99,100].

Deconditioning, loss of confidence, loss of function and isolation from the work team can occur rapidly when the usual activity levels and work participation of soldiers reduce following injury, during the initial treatment phase [101]. It is therefore essential that physical conditioning and progressive return to work commence as soon as the injury is stabilised and concurrently with injury treatment, within the bounds of what is safe and beneficial for the soldier [102,103]. For example, a soldier who has experienced a tibial stress fracture may be prohibited from full weight bearing for some time to unload the bone and allow healing, but assuming the stress fracture is stable and they are not required to wear a plaster cast, they may be able to participate in ‘water running’ in deep water (eliminating weight bearing through the injured leg) during this time, in order to maintain aerobic fitness levels, muscle function and joint movement [101]. They may also be able to participate in strength training, seated or in other positions that do not require weight bearing through the injured leg. In addition, they may be able to undertake a range of usual military tasks, which do not require full weight bearing through the injured leg, in familiar work environments and supported by peers and supervisors. The aim is always to maintain as much conditioning and normal functioning and interaction in the work environment as possible, while at the same time protecting the healing process for the injury [99,100,104].

Once the initial tissue healing phase is complete and the soldier is ready to return to, and progressively rebuild their capacity, for weight-bearing activities consideration should be given to preparation and reconditioning for load carriage and other military tasks. As with most forms of conditioning and return to work rehabilitation, specificity is of importance and concepts such as the ‘principle of specificity’ [105] and ‘Specific Adaptation to Imposed Demands’ [105] should form part of preparing and reconditioning soldiers for load carriage and other military tasks. Research shows that load carriage-specific training is optimal to improve load carriage performance [106,107,108]. Furthermore, Rudzki [109] compared groups of military recruits assigned to either a running platoon or a load marching platoon, and the load marching group were subjectively rated by staff as performing better at military tasks overall than the run group.

Given that load carriage itself constitutes a source of injury risk, the frequency with which load carriage training is scheduled during preparation or reconditioning following injury is of importance. Research suggests that a load carriage-specific session should be conducted at least once every seven to 14 days [106,107]. However, with an increased risk of injury and no additional improvements in load carriage performance found when sessions were greater than four per month [110], the recommendation is that load carriage-specific sessions are conducted no more than once every 10 to 14 days, since a further increase in frequency will typically not increase performance but may increase the risk of injury [30]. This reasoning is also strengthened by the observation of a slow recovery of neuromuscular function in the trunk and limbs up to 48–72 h after a load carriage bout [11,111,112]). For example, in a study by Leyk et al. [112], military ambulance workers performed a maximal stretcher transport with a mean load of 25 kg on each side and wearing 10 kg of standard military clothing. Following this task, grip strength was still significantly reduced 72 h later, leading the authors to assume that the vertical movements of the stretcher led to eccentric stress and to muscle damage during the maximal stretcher transport.

As part of a load carriage conditioning program, intensity and volume need to progressively increase, at a safe rate, to meet the occupational requirements of the soldier. Considering this, while volume can be manipulated by changing the duration or distance of a load carriage event, intensity can be more adequately controlled by manipulating the speed of march, grade, and type of terrain. Exploiting these factors may be particularly useful in the early stages of reconditioning following injury, when loads carried, speeds of movement, or forces applied to specific body parts need to be limited. For example, if the rehabilitation focus was on increasing weight bearing through the recovering body site of injury but current aerobic fitness was limited, load mass could be increased from 15 to 20 kg, while the speed of march could be decreased (5.0 km/h to 4.5 km/h). This example demonstrates how, while relative energy costs and workloads can be kept similar, gradual increases to actual load mass can be made. Alternatively, if the rehabilitation focus was to increase load carriage-specific aerobic fitness but no additional load mass was to be added to the recovering body site, the load mass could be kept constant (e.g., 15 kg), while the surface grade could be increased (0 to 3%) or the terrain swapped from formed roadway to soft sand, thereby maintaining load mass but increasing energy costs and workload to facilitate physical conditioning while avoiding overload of specific body sites.

While load carriage-specific sessions, limited to every 10 to 14 days, are an essential component of load carriage conditioning and reconditioning programs, so too are resistance and aerobic training [106,107]. A combination of resistance and aerobic training has been associated with improvements in load carriage performance [106,107,108]. Studies show that upper body relative strength (strength per unit of body mass) is more highly correlated to loaded road march performance than lower body relative strength [108,113]. Furthermore, increases in aerobic and musculoskeletal fitness are likely essential to prevent future injury, with lower levels of these measures associated with an increased risk of injury in soldiers [114,115,116]. As such, resistance training (notably relative strength based) and aerobic conditioning should form part of the load carriage conditioning/reconditioning plan for soldiers required to carry loads [107].

Factors external to the load carriage program need to be considered. If the soldier is to remain with their unit whilst undergoing conditioning or reconditioning for load carriage, the impacts of their other duties need to be considered and where necessary reduced to aid in prevention of injury or reinjury [117]. Program-Induced Cumulative Overload (PICO) is a term used to highlight the impacts that other duties and tasks of a physical nature can have on a soldier undertaking a physical conditioning program [118]. As an example, soldiers may be required to complete basic weapon skill training wearing full loads or spend a period of time marching on a parade ground prior to a load carriage conditioning session [118]. These additional musculoskeletal loads may pre-fatigue the soldier, possibly increasing their risk of injury during any subsequent planned load carriage session. Other examples include informal distances covered by soldiers simply moving between lessons, the mess, the barracks, etc., with these distances often equating to 7–11 km per day in some situations [119,120] and further increasing the total training load. On this basis, daily program factors other than load carriage training need to be considered if the load carriage conditioning or reconditioning plan is to be suitably progressive without leading to excessive overload. To mitigate some of these concerns, consideration of the program could allow for load carriage conditioning to be completed as part of other programmed activities. For example, loads worn during a 40 min weapon training session could constitute the load carriage conditioning session for a particular 10–14 days period, with progression achieved by including marching to and/or from the weapon training session with loads. Likewise, portions of the soldier’s day, once every 10–14 days, could involve wearing load while undertaking programmed military training or administration, thereby providing a loading benefit when the soldiers move around their military area during the day.

While this paper has focussed on the physical aspects of load carriage, a final consideration is the impact of mental acuity associated with load carriage. The psychological impacts of physical injury on return to work and performance, and impacts of an injury on mental health are beyond the scope of this paper. However, it should be noted that load carriage is known to impact on aspects of mental acuity, such as attention to task [13]. As such, adding cognitive challenges to the rehabilitation of personnel while conducting load carriage tasks (e.g., remembering number sequences, and identifying the number of ‘red circles’ or ‘blue squares’ on a marching route) could be beneficial. Likewise, the addition of military psychologists to the allied health team may usefully inform the overall rehabilitation process.

## 7. Conclusions

With soldiers required to carry increasingly heavy external loads that have potential to cause injury, optimal conditioning and reconditioning (following injury) practices are of importance to both the individual and the army in which they serve. When implemented, load carriage conditioning should include an appropriate load carriage-specific session every 10–14 days, with training progressions achieved through manipulation of load weight, speed, distance, and grade and type of terrain. The load carriage conditioning program should also include aerobic fitness and resistance training sessions, while considering factors external to the load carriage program which can impart a physical load, such as other military duties and training (particularly those including load carriage).

## Data Availability

Not applicable.

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
