# Peer review of "Soldier Load Carriage, Injuries, Rehabilitation and Physical Conditioning: An International Approach"

_ijerph, 2021, doi:10.3390/ijerph18084010_

Round 1

Reviewer 1 Report

The article presents a problem of great interest to researchers and professionals involved in the army.

The Weight of the Soldier’s Load 

  • Another big difference between these professionals is the time and distances they have to travel with the loads, isn't it?

Physiological Responses to Load Carriage 

  • It is true that variables such as speed, grade or terrain type increase intensity, but you have previously commented that it was the load carrying which injured soldiers, not the speed, grade or type of terrain, so I do not understand what "direct" use these variables can have in a reconditioning programme.

Conclusions 

  • The authors review the subject, commenting on some of the important aspects to be taken into account in military injuries, but they do not make any valuable contribution and the conclusions are very poor.

Author Response

Reviewer 1:

The article presents a problem of great interest to researchers and professionals involved in the army.

The Weight of the Soldier’s Load 

  • Another big difference between these professionals is the time and distances they have to travel with the loads, isn't it?
    • Thank you. Correct. We have added a sentence to that effect. Lines 75-80.

Physiological Responses to Load Carriage 

  • It is true that variables such as speed, grade or terrain type increase intensity, but you have previously commented that it was the load carrying which injured soldiers, not the speed, grade or type of terrain, so I do not understand what "direct" use these variables can have in a reconditioning programme.
    • Thank you. Yes it was load carriage but that as noted (and commented by yourself above) load carriage encompasses not only the load weight but the context in which it is carried. As also noted in the research by Orr, et al. (Load carriage: Minimising soldier injuries through physical conditioning-A narrative review. J Mil Veterans Health 2010, 18, (3), 31-38), optimised load carriage conditioning includes aspects of Frequency, Intensity (made up of load and speed and terrain), Time / Duration and Type of training (being load carriage).
    • Furthermore, as also noted aerobic fitness is a leading requirement for load carriage, as such, rather than just adding load to an injured knee (as it returns to fully loaded weight bearing status) to increase load carriage aerobic specific fitness, the speed of march can be increased, or the incline can be increased. Conversely, if the focus is on adding weighted load to a recovering knee but the recovering soldier still has a low aerobic fitness, load weight can be added, and the speed of march and terrain can be adjusted to reduce aerobic costs.
    • We have reworded this section (lines 310-323)

Conclusions 

  • The authors review the subject, commenting on some of the important aspects to be taken into account in military injuries, but they do not make any valuable contribution and the conclusions are very poor.
    • Respectfully we disagree. This review is written specifically for those involved in the conditioning and rehabilitation of military personnel. There is no known synthesis of the literature as complete as this one on soldier load carriage for both conditioning and importantly rehabilitation (from injury through to full return to work) of soldiers given from the viewpoint of authors who are not only researchers in this field (and have written the majority of the current reviewers since 2004) but are also responsible for this work across seven different countries. What is needed is an easy to understand and follow evidence-based approach that a physical therapist rehabilitating military personnel or physical training instructors and strength and conditioning coaches training military personnel can use to inform their treatment.
  • We would like to thank the reviewer for their time and efforts to improve this manuscript.

Reviewer 2 Report

The article was written in accordance with the principles of narrative review.
The manuscript has some drawbacks to which I would like to draw the authors' attention. Well, the title may contain abstract elements and should reflect the essence of the topic, but it should be attractive enough to encourage the reader to read the summary itself, and then the entire article. Particular attention should be paid to the form of the title and keywords that are used by the database to index the article.
I suggest changing the title to a more attractive one. For example: biomechanical responses, physical condition and rehabilitation related to injuries in a soldier's external stress assignment: an international approach. As for the content of the article itself, it is an original look at the issues of injuries and factors influencing the effectiveness of the operation of soldiers burdened with external loads.
The proposal is not very ambitious. The authors point out the need to extend aerobic and resistance training. It seems obvious that in addition, hard training in the gym should be considered. For researchers who want to take up the research problem regarding the burden on soldiers in the future, this article should be helpful in constructing such a work and carrying out the research process. This article can stimulate scientific dialogue and support future research.

Author Response

Reviewer 2:

The article was written in accordance with the principles of narrative review.
The manuscript has some drawbacks to which I would like to draw the authors' attention.

Well, the title may contain abstract elements and should reflect the essence of the topic, but it should be attractive enough to encourage the reader to read the summary itself, and then the entire article. Particular attention should be paid to the form of the title and keywords that are used by the database to index the article. I suggest changing the title to a more attractive one. For example: biomechanical responses, physical condition and rehabilitation related to injuries in a soldier's external stress assignment: an international approach.

  • Thank you for the suggestion However, we fell our existing title is very appropriate to the scope of the article but agree the quantify ‘physical’ should be added. As such the title has been amended to ‘Soldier Load Carriage, Injuries, Rehabilitation and Physical Conditioning: An International Approach’

As for the content of the article itself, it is an original look at the issues of injuries and factors influencing the effectiveness of the operation of soldiers burdened with external loads. The proposal is not very ambitious. The authors point out the need to extend aerobic and resistance training. It seems obvious that in addition, hard training in the gym should be considered.

  • Thank you. While it may seem intuitive that gym training is needed, it likewise highlights that gym training alone will be very limited. However, it also does note the importance of both aerobic training AND strength training. This is of importance as often the mindset is that only load carriage can improve load carriage and that often most of the training is aerobic based (i.e., going for a long run or march) which is easier to do in large groups where equipment is limited. As such, the aim of this article is to inform uniformed physical therapists and uniformed military physical training instructors required to deliver optimised rehabilitation for load carriage with a distilled and evidence-based summation of all the literature though which to guide rehabilitation and conditioning.

Reviewer 3 Report

Review: Soldier Load Carriage, Injuries, Rehabilitation and Conditioning: An International Approach IJERPH 1147315

This review was well written, informative and I found it an interesting and enjoyable read. The English is good. I could find little to fault the authors rationale of why the review was written or the way they conveyed this information to the reader.

An aspect that might have been discussed more fully in this review is whether mental rehabilitation impacts on the recognitioning procedures the authors have suggested. The authors elude to this with their comment “Deconditioning, loss of confidence, loss of function and isolation from the work team”

When an individual is overstressed from carrying an excessive load retrograde effects on mental accuity would also be expected to be associated with aerobic deficiency depending on the overall fitness of an individual impacting on an individuals ability to “think on the run”, clarity of thought and crucial decision making also liable to be detrimentally affected. This is an aspect of the functional recovery of an individual that is not covered in this study but would appear to be an essential component that needs to be considered in order to achieve full functional recovery. Mentally overstressed individuals have an altered energy metabolism with nervous energy imposing a significant drain on efficient utilization of available energy reserves with a decrease in energy available to utilize for muscle activity reflected in aerobic efficiency and endurance capability.

It was once said to me by a famous international rugby union coach that you can improve a persons fitness and recovery from injury by significantly impacting on an individual from the shoulder down but what happens above the shoulder may be even more significant.  The best players tend to respond positively from fitness and skill training – skill training is essential to get the individuals mental faculties on board and focused in a co-ordinated manner with the physical activity exerted by an individual. Having an active mental involvement in rehabilitative and reconditioning procedure is essential to ensure optimal landmarks of aerobic fitness and endurance performance are achieved in a timely manner.

I understand why recovery procedures must be stylized to the needs of a particular individual. A further aspect which might have been explored is how group rehabilitative/reconditioning programmes might be used to improve on these procedures with such group events providing sustained effort from individuals due to the natural competitiveness of the participants in the group which fits in well with the ethos of team effort.

Minor typos

Ref 34 –full details are required of the authors, who was the publisher

The authors should ensure that all required information is provided for all references.

Author Response

Reviewer 3:

For researchers who want to take up the research problem regarding the burden on soldiers in the future, this article should be helpful in constructing such a work and carrying out the research process. This article can stimulate scientific dialogue and support future research.

Review: Soldier Load Carriage, Injuries, Rehabilitation and Conditioning: An International Approach IJERPH 1147315

 This review was well written, informative and I found it an interesting and enjoyable read. The English is good. I could find little to fault the authors rationale of why the review was written or the way they conveyed this information to the reader.

 An aspect that might have been discussed more fully in this review is whether mental rehabilitation impacts on the reconditioning procedures the authors have suggested. The authors elude to this with their comment “Deconditioning, loss of confidence, loss of function and isolation from the work team”

 When an individual is overstressed from carrying an excessive load retrograde effects on mental acuity would also be expected to be associated with aerobic deficiency depending on the overall fitness of an individual impacting on an individuals ability to “think on the run”, clarity of thought and crucial decision making also liable to be detrimentally affected. This is an aspect of the functional recovery of an individual that is not covered in this study but would appear to be an essential component that needs to be considered in order to achieve full functional recovery. Mentally overstressed individuals have an altered energy metabolism with nervous energy imposing a significant drain on efficient utilization of available energy reserves with a decrease in energy available to utilize for muscle activity reflected in aerobic efficiency and endurance capability.

 It was once said to me by a famous international rugby union coach that you can improve a persons fitness and recovery from injury by significantly impacting on an individual from the shoulder down but what happens above the shoulder may be even more significant.  The best players tend to respond positively from fitness and skill training – skill training is essential to get the individuals mental faculties on board and focused in a co-ordinated manner with the physical activity exerted by an individual. Having an active mental involvement in rehabilitative and reconditioning procedure is essential to ensure optimal landmarks of aerobic fitness and endurance performance are achieved in a timely manner.

I understand why recovery procedures must be stylized to the needs of a particular individual. A further aspect which might have been explored is how group rehabilitative/reconditioning programmes might be used to improve on these procedures with such group events providing sustained effort from individuals due to the natural competitiveness of the participants in the group which fits in well with the ethos of team effort.

  • Thank you. We defiantly agree but wanted to tread carefully and not extend too far beyond both the scope of this paper and the qualifications of the authors. We have added a small paragraph which we hope to identify this need.

Minor typos

Ref 34 –full details are required of the authors, who was the publisher

  • Thank you.

The authors should ensure that all required information is provided for all references.

  • Thank you.

Round 2

Reviewer 1 Report

Thank you for clarifying the doubts presented. Congratulations for the work done.